# Demographic and Clinical Predictors of Drug Response in Epileptic Children in Jeddah

**DOI:** 10.3390/biomedicines11082151

**Published:** 2023-07-30

**Authors:** Rania Magadmi, Reem Alyoubi

**Affiliations:** 1Clinical Pharmacology Department, Faculty of Medicine, King Abdulaziz University, Jeddah 21589, Saudi Arabia; 2Pediatric Department, Faculty of Medicine, King Abdulaziz University, Jeddah 21589, Saudi Arabia; raalyoubi@kau.edu.sa

**Keywords:** antiseizure, drug response, pediatric neurology

## Abstract

Epilepsy is a chronic neurological disease of the brain. Over 20 antiseizure medications are available on the market, but a third of patients still have drug-resistant epilepsy. This study was designed to assess the impact of the demographic and clinical characteristics of epileptic children on their likelihood of developing drug resistance. This study was a multicenter, hospital-based, cross-sectional, case–control study of pediatric patients diagnosed with epilepsy in Jeddah, Saudi Arabia. The study included 101 children with epilepsy. Fifty-six patients showed good response to antiseizure medications (ASMs), and forty-five patients had a poor response. A statistically significant good response to ASMs was reported among younger patients, those who did not report parental consanguinity, those who did not have a family history of epilepsy, and those diagnosed with partial seizures, with no reported adverse effects. The levetiracetam regimen was statistically significant regarding the responsiveness to ASMs. Patients on a monotherapy regimen elicited a significantly better response to levetiracetam than patients on polytherapy (*p* < 0.001). No significant association was found between the response to ASMs and the sex, nationality, body mass index, complete blood count, or vitamin B12 level. In conclusion, the ASM response in epileptic patients can be predicted by knowing the patient’s demographic and epileptic history. However, the complete blood count and vitamin B12 level failed to predict patients’ response to ASMs.

## 1. Introduction

Epilepsy is a chronic neurological disorder that affects millions of people worldwide, causing significant medical and social burdens. This condition is particularly prevalent among pediatric patients, leading to an abnormal development and a decreased quality of life. In Saudi Arabia, the prevalence of epileptic patients is estimated to be 6.54 per 1000 [1].

Antiseizure medication (ASM) therapy is considered the standard treatment for epilepsy. Currently, there are more than 20 available ASMs, most of which have been developed and well used for many years. These ASMs differ in their chemical structure and mechanism of action [2]. Despite the wide variety of ASM options available, approximately one-third of epileptic patients are resistant to these medications [3]. A retrospective study conducted in Saudi Arabia found that 30% of adult epileptic patients were ASM-resistant [4]. Hence, the International League Against Epilepsy (ILAE) has defined drug-resistant epilepsy as “a failure of adequate trials of two tolerated, appropriately chosen and used ASM schedules (whether as monotherapies or in combination) to achieve sustained seizure freedom” [5].

ASM resistance can result in psychological disturbance and an impaired quality of life, and can increase the risk of adverse drug reactions (ADRs) among pediatric patients [6]. Moreover, there is increasing concern about the effectiveness and the safety profiles of some ASMs.

There are several demographic factors that may influence ASM resistance in pediatric patients with epilepsy, such as gender and age [6]. The clinical characteristics of epileptic children may also be associated with ASM resistance; for instance, the type of epilepsy. However, there is limited research on the demographic and clinical characteristics of pediatric patients with drug-resistant epilepsy in Saudi Arabia.

Therefore, this study aims to fill in this knowledge gap, and investigate the potential correlation between the demographic and clinical characteristics of epileptic children and ASM resistance in pediatric patients in Jeddah, Saudi Arabia. By understanding these associations, healthcare professionals can identify high-risk populations, implement targeted interventions, and improve the overall management of drug-resistant epilepsy among Saudi Arabian pediatric patients.

## 2. Materials and Methods

### 2.1. Study Design

This study was a multicenter, hospital-based, cross-sectional, case–control study of pediatric patients diagnosed with epilepsy in King Abdulaziz University Hospital (KAUH) and Soliman Fakeeh Private Hospital, during the period from November 2020 to April 2021.

The diagnosis of drug-resistant epilepsy was based on the International League Against Epilepsy (ILAE) guidelines [5]. Poor responders were defined as “Epileptic patients with failure of adequate trials of two tolerated and appropriately chosen and used ASM schedules (whether as monotherapies or in combination) to achieve sustained seizure freedom”. The good responder group was defined as “epileptic patients receiving the current ASM regimen and seizure-free for a minimum of 12 months”. The demographic and clinical data, and the chemical parameters of patients were collected, and compared between cases (poor responders) and controls (good responders).

The inclusion criteria for the study were children < 18 years of age who were diagnosed with epilepsy and had been on one or more antiseizure medications for at least one year. Additionally, the participants should present a normal psychometric development, and a normal neurological examination. The patient’s caregiver had to provide informed consent for the participant to be included in the study. Patients who did not comply with their ASM treatment, had a history of liver disease, did not visit the clinic regularly, had incomplete data, or whose caregiver refused to provide informed consent were excluded.

All the caregivers of the participants in the study were required to give written informed consent after an explanation of the study’s aims and procedures by the pediatric neurologist. The participants were informed that they were free to withdraw from the study at any time. If participants withdrew before giving a blood sample, their data were removed from the sample size. This study was approved by the biomedical ethics research committee of the Faculty of Medicine, King Abdulaziz University, on 26 October 2020 (approval number 530-20).

The initial screening involved 200 patients. On the basis of the inclusion and exclusion criteria, 150 patients were eligible to participate in this study. The participants’ caregivers were asked to read and sign the informed consent form after the primary investigator explained the study’s purpose and process to them. Another co-investigator was a witness, as well. Of these patients, 49 patients refused to complete the procedure program. Of the remaining patients, 101 agreed to participate in this study, and provided complete data.

A designed electronic data sheet was created. The form was used to gather information about the 101 patients’ demographic and clinical data. Demographic data that may affect the ASM response were included, such as age, body mass index, gender, nationality, family history of epilepsy, and parental consanguinity. The clinical data included the type of seizures, and the duration and type of the current ASM. Reported ASM-induced adverse drug reactions (ADRs), such as electrolyte disturbance, liver function test (LFT) disturbance, cognitive or motor delay, and weight changes were also recorded.

### 2.2. Statistical Analysis

The demographic, clinical, and laboratory data of epileptic patients were analyzed using descriptive statistics. The frequencies and percentages were reported for the categorical variables. Continuous data were reported as mean ± standard deviation. Differences between groups were analyzed using Pearson’s chi-square to test categorical variables, and a *t*-test for continuous outcome variables. The significance level (*p*-value) was set at 0.05. The statistical analysis was performed using Social Sciences Statistical Package (SPSS) software version 21 (IBM, Armonk, NY, USA).

## 3. Results

### 3.1. Demographic Characteristics of the Patients

Out of the 101 epileptic patients, 67.3% were male and 65.3% were Saudi. The mean age of the participants was 7.3 years (the minimal value was 2, and the maximal 16), and the mean body mass index was 16.5 (the minimal value was 9, and the maximal was 35). A total of 93% of the cases did not declare a family history of epilepsy, although parental consanguinity was reported among approximately half of the study sample, as shown in Table 1.

### 3.2. Clinical Characteristics of the Patients

As shown in Table 2, the majority of the patients were diagnosed with generalized tonic–clonic epilepsy (45.5%). Around 40% of the patients had been on their medication for more than two years.

Regarding ADRs, electrolyte disturbance was reported in two-thirds of the patients. Liver function test disturbance was reported in 40%, and cognitive or motor delay was reported in <20%. More than half of the patients had a normal body weight (56.4%); however, 20 out of the 101 patients were obese or overweight.

### 3.3. Characteristics of Antiseizure Medications Used by Epileptic Patients

As shown in Table 3, about two-thirds of the participants were on ASM monotherapy at the time of the study, while the one-third of the patients were on ASM polytherapy. The most commonly prescribed drugs were levetiracetam (as monotherapy in 28 patients, and in polytherapy in 25), and valproic acid (as monotherapy in 13 patients, and in polytherapy in 14).

Based on the ASM classification, more than half of the patients showed a good response, while 45.5% were poor responders in this cohort.

### 3.4. Demographic Data as Predictors to Antiseizure Medications Response in Epileptics Patients

The results revealed a significant difference in the age, parental consanguinity, and family history of epilepsy between the patients, as shown in Table 4. A significantly good response to ASMs was reported among younger age groups, those who did not report parental consanguinity, and those who did not have a family history of epilepsy. However, no significant associations were found regarding body mass index, sex, and nationality.

### 3.5. Clinical Parameters as Predictors to Antiseizure Medication Response in Epileptic Patients

The results revealed a significant difference in the types of seizure, duration of treatment, and reported adverse drug reactions, as shown in Table 5. A significantly good response to ASM was reported among those who had partial seizures, and an intermediate duration of treatment, and those who had a good response to monotherapy. A significantly poor response was reported among patients with electrolyte disturbance, liver function test disturbance, and cognitive or motor delay. However, there was no significant association with weight change.

### 3.6. Complete Blood Count Parameters and Vitamin B12 as Predictors to the Response to Antiseizure Medications in Epileptic Patients

Regarding the complete blood count parameters as predictors of ASM response in epileptic patients, the results revealed no significant differences between good responders and poor responders, as shown in Table 6. However, all complete blood count parameters among good responders were slightly higher than among poor responders.

### 3.7. The Effect of Antiseizure Medication Regimen on Epileptic Patients’ Response

A chi-square test was performed to examine the relationship between the drug response and the ASM regimen. The relationship between these variables was significant, as shown in Table 7. Participants on a monotherapy regimen showed a better response to levetiracetam than participants on polytherapy. Similar trends were observed with valproic acid, topiramate, carbamazepine, phenobarbital, and lamotrigine, although they did not reach statistically significant levels.

## 4. Discussion

Epilepsy is a highly prevalent disease that affects almost 50 million individuals worldwide. In Saudi Arabia alone, a community-based study conducted by Al Rajeh et al. in 2001 revealed that there are 6.54 epileptic patients for every 1000 individuals [7]. The figure indicates the significant health concern that epilepsy poses, and the need for proper attention and treatment. One alarming aspect of epilepsy is the increased risk of premature death among epileptic patients, compared to the general population. Milroy’s study in 2011 found that the risk can be up to three times higher [8]. Hence, it is evident that epilepsy is not just a medical condition that causes seizures, but also significantly impacts people’s lives and wellbeing.

Many factors could affect patients’ responses to antiseizure medications (ASMs). The early prediction of a patient’s response to ASMs could save the patient time and health. Thus, the current study aimed to predict which patients would respond well to ASMs, based on their demographic, clinical, and laboratory data. The early prediction of patient response to ASMs is crucial, in order to save time and ensure the best possible health outcomes.

One alarming fact is that nearly 75% of epilepsy cases start in childhood. This reveals that children’s developing brains are more vulnerable to seizures than adults’ fully developed brains. Additionally, the triggers for seizures in children are often different from those in adults, further increasing the severity of childhood epilepsy [9]. This highlights the urgent need for early intervention and effective management strategies in pediatric epilepsy cases.

The results from the current study found that 45% of the participants had drug-resistant epilepsy. This percentage is alarmingly higher than the drug-resistant epilepsy incidence reported in prior studies [10,11]. Similarly, the results align with the DRE rates prevalent in Jordanian epileptic children [12]. Additionally, poor response rates of 44–46% were reported in two previous studies conducted among Indian populations [13,14]. The consistent occurrence of drug-resistant epilepsy within the same ethnic group emphasizes the importance of demographic and genetic variables in predicting drug-resistant epilepsy. These factors must be thoroughly evaluated and accounted for in epilepsy management strategies, for the best possible outcomes.

The demographic results from the current study showed that patients become more susceptible to a poor response to ASMs with age. This supports previous studies reporting that the age of seizure onset is significantly lower in the resistant groups [15,16]. A plausible explanation for this trend is that the pharmacokinetics of ASMs can vary with age, as proposed by Zhao et al. [17]. Their study revealed that drug clearance (adjusted for body weight) was lower in adult patients than in younger children, resulting in reduced ASM plasma levels. The other demographic predictors of ASM responsiveness in the current study were the absence of parental consanguinity, or a family history of epilepsy. These results emphasize the importance of considering patient demographics when selecting appropriate treatment options.

Interestingly, the results revealed that the clinical presentation and data of patients can predict the ASM drug response. There were strong correlations between the type of epilepsy, the duration of the disease, and the development of adverse drug reactions with ASM pharmaco-resistance. This supports a previous prospective study with 478 epileptic patients [18]. Three clinical predicting factors for ASM pharmaco-resistance were identified: the type of epilepsy, the duration of the epilepsy, and the number of seizures at the time of diagnosis.

Regarding the epilepsy type, generalized tonic–clonic epilepsy was predominant among poor responders (65.2% vs. 29.1%). This supports some previous research [19,20,21]; however, other studies have demonstrated the opposite [15,22], due to changes in the definition of epilepsy classifications that take into account the type of onset seizures [15,23].

The results from this study show that patients who have been on ASMs for longer have been observed to exhibit a poorer response, compared to those who have been on ASMs for a shorter time. This could be attributed to the development of drug-resistant epilepsy, which may occur over time, as the brain adapts to the medication. One possible explanation for this phenomenon is the down-regulation or decreased expression of drug targets in the brain, reducing the efficacy of the medication over time. Additionally, chronic exposure to ASMs may lead to an increased drug metabolism, resulting in a reduced drug concentration, and diminished therapeutic effects [5].

Can et al. (2020) reported that the most-used antiseizure medications among the study population were valproate, levetiracetam, and lamotrigine [24]. Similarly, in the current study, levetiracetam, valproate, and topiramate were the most-used antiseizure medications. Furthermore, it was observed that ASM drug resistance was common in patients treated with more than two ASM therapies, while drug-responsive patients responded better to monotherapy treatment. In particular, the association between the drug regimen and ASMs was more significant among patients on an ASM regimen that included levetiracetam. Ajmi et al. (2018) and Can et al.’s (2020) results support this [15,25]. In the study, all the ASMs showed different response rates. Phenobarbital, levetiracetam, and valproic acid were the most effective drugs as monotherapy. A study by Gesche (2002) found that valproic acid was the most effective drug among adult epileptic patients [25].

Vitamin B12 levels were investigated as a potential factor affecting the metabolism of ASMs and, subsequently, their therapeutic effect [26]. Previous research has shown that the chronic use of ASMs resulted in a decrease in the serum levels of vitamin B12, and disturbances in the complete blood-count parameters [27]. Indeed, individual genes, the epilepsy type, the medication used, and the dose of ASMs are all factors that can affect the level of vitamin B12 [28]. A study by Huang et al. (2016) found that the long-term administration of ASMs affected the metabolism of serum vitamin B12, and led to an increased risk of neurological and cardiovascular disorders [27]. This suggests that vitamin B12 could be an independent risk factor for the ASM response.

Some ASMs are enzyme inducers, such as phenytoin and carbamazepine, and directly affect the activity of variant liver enzymes. Liver enzyme induction can cause the depletion of cofactors, such as vitamin B12, which leads to altered homocysteine levels [29]. The methylated forms of vitamin B12 can cause problems related to methylenetetrahydrofolate reductase (MTHFR) gene handling. Nevertheless, a hereditary predisposition to vitamin B12 deficiency has been observed in different studies. For example, a study in Jordan [30] demonstrated a significant correlation between 677CT of the MTHFR gene mutation, and vitamin B12 deficiency in the study population. The results from this study found that the complete blood-count parameters and vitamin B12 level alone could not predict the patient response.

## 5. Conclusions

The results of this study suggest that there may be a potential relationship between demographic and epileptic history, and the prediction of the antiseizure medication (ASM) response, in patients with epilepsy. These findings imply that a thorough patient diagnosis and classification process could be vital in personalizing the choice of ASM medications. However, it should be noted that the prediction of the ASM response based on complete blood-count parameters and vitamin B12 levels did not yield significant results in the current study. However, the complete blood count parameters should be checked periodically in patients receiving ASMs, as some medication leads to a disturbance in their blood component, and this can be more aggressive in patients with a genetic mutation.

To gain a more comprehensive understanding of the relationship between epilepsy and the ASM response, further investigations are warranted. These would preferably involve a larger sample size, and the examination of the additional gene polymorphisms that contribute to a vitamin B12 predisposition in individuals.

These suggested findings provide valuable insights into the potential factors influencing the ASM response in epileptic patients. However, it is crucial to exercise caution when interpreting the results, as further research and corroborating evidence are necessary to fully establish these associations.

## Figures and Tables

**Table 1 biomedicines-11-02151-t001:** Demographic characteristics of the patients.

Character	Mean (SD)	
Age	7.3 (4.1)	
BMI	16.5 (4.5)	
Character		N (%)
Sex	Male	68 (67.3%)
	Female	33 (32.7%)
Nationality	Saudi	66 (65.3%)
	Non-Saudi	35 (34.7%)
Parental consanguinity	Yes, first degree	33 (32.7%)
	No	68 (67.3%)
Family history of epilepsy	Yes	7 (6.9%)
	No	94 (93.1%)

Data are presented as mean (±SD) or as number (%). BMI, body mass index.

**Table 2 biomedicines-11-02151-t002:** Clinical characteristics of the patients.

	Character	N (%)
Types of seizure	Generalized tonic–clonic epilepsy	46 (45.5%)
	Generalized myoclonic epilepsy	21 (20.8%)
	Partial seizure	34 (33.7%)
Duration of treatment	1 year–2 years	61 (60.4)
	>2 years	40 (39.6)
ADR	Electrolyte disturbance	63 (62.3%)
	LFT disturbance	41 (40.6%)
	Cognitive vs. motor delay	20 (19.8%)
Weight change	Healthy weight	57 (56.4%)
	Obese	12 (11.9%)
	Overweight	8 (7.9%)
	Underweight	24 (23.8%)

Data are presented as mean (±SD) or as number (%). ADR, adverse drug reaction; LFT, liver function test.

**Table 3 biomedicines-11-02151-t003:** The characteristics of the antiseizure medications used by the epileptic patients.

	Character	N (%)
Antiepileptic drug regimen	Monotherapy	62 (61.4)
	Polytherapy	39 (38.6)
Antiepileptic drugs (alone or in combination)	Levetiracetam	53 (52.5)
	Valproic acid	27 (26.7)
	Topiramate	20 (19.8)
	Carbamazepine	17 (16.8)
	Phenobarbital	12 (11.9)
	Lamotrigine	5 (5)
Patient classification based on the ASM drug response	Good responder	55 (54.5)
	Poor responder	46 (45.5)

Data are presented as mean (±SD) or as number (%). ASM, antiseizure medication.

**Table 4 biomedicines-11-02151-t004:** Demographic data as predictors of the response to antiseizure medication in epileptic patients.

Character		Good Responders N (%)	Poor Responders N (%)	*p*-Value
N		55 (54.5)	46 (45.5)	
Age #		6.2 (3.4)	8.5 (4.6)	0.006 *
BMI #		16.5 (3.8)	16.7 (5.3)	0.829
Sex ^	Male	37 (67.3)	31 (67.4)	0.99
	Female	18 (32.7)	15 (32.6)	
Nationality ^	Saudi	40 (72.7)	26 (56.5)	0.088
	Non-Saudi	15 (27.3)	20 (43.5)	
Parental consanguinity ^	Yes, 1st degree	12 (21.8)	21 (45.7)	0.039 *
	No	43 (78.2)	25 (54.3)	
Family history of epilepsy ^	Yes	2 (3.6)	5 (10.9)	0.008 *
	No	53 (96.4)	41 (89.1)	

Data are presented as mean or as number (%). BMI, body mass index. # Comparison was performed using an independent *t*-test. ^ Comparison was performed using *chi*-square. * *p* value < 0.05 considered significant.

**Table 5 biomedicines-11-02151-t005:** Clinical parameters as predictors of the response to antiseizure medications in epileptic patients.

	Character	Good Responders N (%)	Poor Responders N (%)	*p*-Value
**N**		55 (54.5)	46 (45.5)	
Types of seizure ^	Generalized tonic–clonic epilepsy	16 (29.1)	30 (65.2)	0.005 *
	Generalized myoclonic epilepsy	16 (29.1)	5 (10.9)	
	Partial seizure	23 (41.8)	11 (23.9)	
Duration of treatment ^	1 year–2 years	47 (85.5)	14 (30.4)	0.0001 *
	>2 years	8 (14.5)	32 (69.6)	
ADR #	Electrolyte disturbance	22 (40)	41 (89.1)	0.0001 *
	LFT disturbance	11 (20)	30 (65.2)	0.0001 *
	Cognitive vs. motor delay	5 (9.1)	15 (32.6)	0.003 *
Weight change ^	Underweight	9 (16.4)	15 (32.6)	0.139
	Healthy weight	33 (60)	24 (52.2)	
	Overweight	13 (23.6)	7 (15.2)	

Data are presented as mean (±SD) or as number (%). ADR, adverse drug reaction; LFT, liver function test. # Comparison was performed using an independent *t*-test. ^ Comparison was performed using *chi*-square. * *p* value < 0.05 considered significant.

**Table 6 biomedicines-11-02151-t006:** Complete blood count parameters and vitamin B12 as predictors of the response to antiseizure medications in epileptic patients.

Parameters	Normal Range	Good Responder Mean (SD)	Poor Responder Mean (SD)	*p*-Value
Hb g/dL	12–15	11.3 (1.9)	10.9 (1.8)	0.31
RBC m/µL	4 –5.2	4.6 (0.6)	4.4 (0.6)	0.07
MCH pg	32–36	26.1 (4.5)	25.3 (4.1)	0.38
HC %	35–49	35.1 (5.2)	34 (4.4)	0.26
WBC (k/µL)	4.5–13.5	8.7 (3.2)	9 (4.7)	0.67
Neutrophil (%)	35–65	45.1 (18)	42.9 (16.8)	0.55
Lymphocytes (%)	10–15	38.4 (18.9)	42.4 (17.5)	0.27
Monocytes (%)	2–11	7.6 (3.8)	8.3 (4.9)	0.39
Eosinophil (%)	1–4	2.2 (2.6)	2.4 (3.2)	0.75
Platelets	150–450	330.9 (98.6)	345.1 (106.6)	0.49
Vitamin B12 (pg/mL)	197–771	84 (19)	83 (16)	0.82

Data are presented as mean (SD). Abbreviation Hb, hemoglobin; RBC, red blood cell; MCH, mean corpuscular hemoglobin; HC, hematocrit; WBC, white blood cell. *p* value < 0.05 considered significant using an independent *t*-test for comparison.

**Table 7 biomedicines-11-02151-t007:** The effect of the antiseizure medication regimen on the response in epileptics patients.

ASM	Regimen	Good Responder N (%)	Poor Responder N (%)	Adjusted OR (95% CI)	*p*-Value
**N**		55 (54.5)	46 (45.5)		
Levetiracetam (N = 53)	Monotherapy (N = 30)	21 (70)	9 (30)	8.4 (2.36 to 28.1)	<0.001 *
	Polytherapy (N = 23)	5 (21.7)	18 (78.3)		
Valproic acid (N= 27)	Monotherapy (N = 13)	9 (69.2)	4 (39.8)	4.05 (0.871 to 17.9)	0.13
	Polytherapy (N =14)	5 (35.7)	9 (64.3)		
Topiramate (N = 20)	Monotherapy (N = 3)	2 (66.7)	1 (33.3)	32 (1.66 to 475)	0.05
	Polytherapy (N = 17)	1 (5.9)	16 (94.1)		
Carbamazepine (N = 17)	Monotherapy (N = 8)	4 (50)	4 (50)	3.5 (0.427 to 23.1)	0.33
	Polytherapy (N = 9)	2 (22.2)	7 (77.8)		
Phenobarbital (N = 12)	Monotherapy (N = 4)	4 (100)	0 (0)	NA	0.06
	Polytherapy (N = 8)	2 (25)	6 (75)		
Lamotrigine (N = 5)	Monotherapy (N = 3)	0 (0)	3 (100)	NA	>0.99
	Polytherapy (N = 2)	0 (0)	2 (100)		

Data are presented as the number of patients (N) and percentage (%). OR, odds ratio; CI, confidence interval; NA, not applicable; ASM, antiseizure medication. Data were analyzed using the *chi*-square test. The OR was estimated by logistic regression analysis, after adjusting for the regimen. * *p* value < 0.05 considered significant.

## Data Availability

The authors confirm that the data supporting the findings of this study are available within the article. The raw data that support the findings of this study are available from the corresponding author upon reasonable request.

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
