# Peer review of "Demographic and Clinical Predictors of Drug Response in Epileptic Children in Jeddah"

_biomedicines, 2023, doi:10.3390/biomedicines11082151_

Round 1

Reviewer 1 Report

This manuscript describes demographic and clinical predictors of response to antiepileptic drugs in Saudi Arabia. The study itself is interesting; however, there are some questions about interpretation of the results. If the authors can well answer the questions, this manuscript should be accepted.

1)      Table 1. The authors used the term “gender”. This term is not biological but sociological one. The authors should use “sex” in place of “gender”.

2)       Table 5. The shorter the duration of treatment was, the better the result was. Simply, does this mean that good responders can stop the treatment early?

3)      Table 6. Vitamin B12 concentration did not show significant difference between good responders and poor responders; however, the authors performed long discussion. This is nonsense.

4)      Table 7. Monotherapy showed better response than polytherapy. Simply, does this mean that good responders do not need polytherapy?

5)      Conclusion. This study did not show usefulness of proper diagnosis and personalization of their medication. In addition, this study did not show necessity of checking CBC during treatment by antiepileptic drugs

Author Response

We thank you for the valuable and constructive comments. We have carefully reviewed the comments, and the manuscript was revised according to the suggestions provided by you. Our detailed responses are in the attached file. We have marked the modifications we made in yellow. The revised version of the manuscript has been approved by all authors.

We hope that the revised version is suitable for publication. We look forward to hearing from you, and we would be happy to respond to any other question or comment that you may have.

Reviewer 2 Report

The role can be interesting.

some clarifications, cite: doi: 10.3390/jcm10102080

Regarding statistics, things I wonder about, are the data important when the sample is very small? that is, in the case of Family history of epilepsy, for example, a very low p is obtained, despite the fact that the percentage is similar. I dont believe it.

  As for the Weight change, I think that 4 categories are not needed, I propose to reduce it to 3 and redo the analysis.

Regarding Types of Seizure, the value of the corrected residual is necessary, if not, this cannot be interpreted between categories, there are differences.

In Table 7, I don't see a clear statistic with such small sample values...

The conclusions must be more humble, nothing is confirmed, it is suggested.

Author Response

(The authors gave the same response as above.)

Reviewer 3 Report

After reading the manuscript my major concerns are as follows:

  1. Now, the Antiepileptic Drugs (AEDs) are defined as the Antiseizure Medications (ASMs). Please, replace the acronym throughout the manuscript.
  2. Please, indicate the study period when the children were examined. Was it a one-year study ?
  3. How many patients were primarily selected for the study? How many children were excluded from this study? What was the proportion of excluded epilepsy children in the total number of children with epilepsy in Saudi Arabia?
  4. Table 1. Please, add minimal and maximal values for Age and BMI of children. Mean values are not enough for this type of research.
  5. Page 2 lines 55-56: The control group is defined as “epileptic patients receiving the current ASM regimen and seizure free for a minimum of 12 months”. In light of this definition how the children were included if their treatment were conducted for at least 3 months (lines 59-60 – inclusion criteria)?
  6. Page 3, line 90: How to use the principal definition (a minimum of 12 months of seizure freedom) in patients with disease duration less than 6 months? It was a 16.8% of such patients with diagnosis of epilepsy less than 6 months.
  7. Page 3: What ASM classification was used to differentiate the patients as good vs. poor responders. Was it monotherapy with ASM? Please, provide more information about classification of “good” and “poor” responders. Which criteria were mainly used?

No comments

Author Response

(The authors gave the same response as above.)

Round 2

Reviewer 1 Report

This manuscript is well corrected according to the reviewer's opinion. The reviewer recommends accepting as is.

Author Response

The reviwer kindly accepted all the modifications.

Reviewer 2 Report

the changes made are minimal. I understand that comparative data should not appear when the sample is minimal.

Otherwise the article is correct.

Author Response

(The authors gave the same response as above.)

Reviewer 3 Report

No further comments

Author Response

(The authors gave the same response as above.)
